# Mercury Exposure and Its Health Effects in Workers in the Artisanal and Small-Scale Gold Mining (ASGM) Sector—A Systematic Review

**DOI:** 10.3390/ijerph19042081

**Published:** 2022-02-13

**Authors:** Kira Taux, Thomas Kraus, Andrea Kaifie

**Affiliations:** Institute for Occupational, Social and Environmental Medicine, Medical Faculty, RWTH Aachen University, Pauwelsstraße 30, 52074 Aachen, Germany; kira.taux@rwth-aachen.de (K.T.); tkraus@ukaachen.de (T.K.)

**Keywords:** work, health, disease, intoxication, heavy metal, neuro-psychological disorders

## Abstract

Gold is one of the most valuable materials but is frequently extracted under circumstances that are hazardous to artisanal and small-scale gold miners’ health. A common gold extraction method uses liquid mercury, leading to a high exposure in workers. Therefore, a systematic review according to the PRISMA criteria was conducted in order to examine the health effects of occupational mercury exposure. Researching the databases PubMed^®^, EMBASE^®^ and Web of Science^TM^ yielded in a total of 10,589 results, which were screened by two independent reviewers. We included 19 studies in this review. According to the quantitative assessment, occupational mercury exposure may cause a great variety of signs and symptoms, in particular in the field of neuro-psychological disorders, such as ataxia, tremor or memory problems. However, many reported symptoms were largely unspecific, such as hair loss or pain. Most of the included studies had a low methodological quality with an overall high risk of bias rating. The results demonstrate that occupational mercury exposure seriously affects miners’ health and well-being.

## 1. Introduction

### 1.1. Gold Mining

Since ancient times, gold has been one of the most desired and noble elements in the world, with an outstanding variety of application areas. One of the most impressive examples for its use in art is the world-famous death mask of the Egyptian pharaoh Tutankhamun. The jewellery industry processes gold for all conceivable kinds of products. Furthermore, gold is also of considerable relevance as a financial reserve for national banks. Germany, for instance, possessed about 3400 t of gold ingots in 2020 [1]. Hence, the gold price has shown an overall upward trend over the last five decades and amounted to approximately 1800 USD/fine troy ounce in 2020 [2,3], indicating it to remain a promising market.

The total annual gold production is, according to the U.S. Geological Survey, about 3300 t in 2019 [4], of which artisanal and small-scale gold mining (ASGM) is estimated to account for 380–450 t per year [5]. As defined by the United Nations (U.N.), artisanal and small-scale gold miners (ASG miners) are persons who engage in mining as “individual miners or [in] small enterprises with limited capital investment and production” [6]. Nevertheless, official figures on the exact number of labourers are, to our knowledge, not obtainable, but approximate values suggest that about 16 million people work as ASG miners [5]. This form of gold mining is predominantly practised in countries of the global south, for example, in Ghana, Ecuador and Indonesia [5].

Gold mining, and ASGM in particular, is of great importance for low- and middle-income countries where it reveals both positive and negative aspects. On the one hand, a striking positive aspect is its economic weight, for which Ghana is presented as an example: the gold mining industry alone amounted to 7.1% of Ghana’s national gross domestic product (GDP) in 2019 (provisional data) [7] with a general increase in revenues [8]. In addition, the companies that are members of the Ghana Chamber of Mines also support society by, for example, financially sponsoring projects in the fields of health or education [8]. On the other hand, negative effects concern the environment [9,10,11,12], with parts of the tropical rainforest in South America being destroyed in the context of gold mining [9]. Mining waste including chemicals such as cyanide or mercury can pollute the environment, such as water, sediments and soil, finally affecting the human food chain, as well [10,11,12]. In addition, the workers’ health is harmed by their work itself. Nakua et al. observed that ASG miners have a high risk of being injured at work, while the safety precautions are at a low standard [13]. Mercury exposure is known to cause a considerable burden of disease in miners, being responsible for up to more than 2 million DALYs per year, especially in countries of the global south [14].

In general, the mining process can be carried out using various methods, depending on available materials, equipment and knowledge [15,16]. Among other methods, the mercury amalgamation is still used for extracting gold [15,16]. Here, gold-containing rocks are ground and afterwards pulverized into small pieces, and the material thus obtained is mixed with elemental mercury to form an amalgam out of gold and mercury [15,16]. After gathering, this gold amalgam is further processed by smelting, so that the mercury vaporizes and the gold remains behind [15,16]. 

### 1.2. Mercury

Mercury (also known as Hg or quicksilver) is a chemical element with a silver–grey colour. It is the only metal that exists in a liquid aggregate state under standard conditions and evaporates on contact with the ambient air. According to an official European Union directive, mercury is categorized as a threat to aquatic ecosystems, as toxic through inhalation and as hazardous to human’s health [17].

Natural processes, such as volcanism, cause atmospheric mercury emissions. However, anthropogenic sources, such as coal combustion or ASGM contribute to at least three-fold higher mercury masses in the atmosphere [18,19,20]. Currently, the ASGM sector is the largest contributor to anthropogenic—man-made, non-natural—mercury emissions [20]. In 2015, its airborne emissions amounted up to 838 t, which represents approximately 38% of the total mercury emissions worldwide [20].

The consequences of mercury exposure are highly dependent on its chemical form (organic or inorganic compounds) as well as its dose and exposure pathway. Mercury can be absorbed via different pathways. ASG miners heat the amalgam to extract gold, leading to an evaporation of metallic (inorganic) mercury [15,16]. Therefore, this paper puts the emphasis on the metallic form. The main absorption route of elemental mercury is via inhalation. In an experimental setting, exposed individuals absorbed 67–87% of the entire inhaled mercury vapour [21]), while absorption of vapour through skin contact and gastrointestinal absorption of ingested metallic mercury played only a subordinate role [22,23]. The exhalative half-life of mercury amounts approximately 2 days [24,25], while the half-life for the urinary excretion is 63 days [25]. However, mercury accumulation occurs in organ tissues, but its exact distribution depends on its chemical form [26,27]. After inhalation of inorganic mercury, the distribution takes place via blood as it crosses most cell membranes, such as the blood–brain barrier or the placenta. The oxidation of elemental mercury in the erythrocytes influences the uptake in the brain with its corresponding typical mercury-related signs and symptoms. In contrast, occupational exposure to organic mercury compounds can be found exemplarily in the production of mercury fulminate. Organic mercury is highly lipophilic and is mainly absorbed via skin and inhalation. The main target organ of organic mercury compounds is the brain. 

Internal human mercury burden can be detected in certain biological materials such as blood or urine, but due to the varying specificity for the different forms of Hg, other materials are suitable for the measurements, such as hair [28]. 

Acute cases of mercury poisoning can occur after contact with its metallic form [29,30,31]. Due to the inhalation of toxic fumes, the resulting symptoms affect, in particular, the pulmonary system with cough and dyspnoea up to acute respiratory distress syndrome (ARDS) [29,30,31]. Additionally, mercury exposure can lead to other unspecific symptoms, such as nausea, diarrhoea, fever or lymphadenopathy [30,31]. The course of mercury poisoning can end lethal, depending on its severity [31]. Likewise, chronic exposure to elemental Hg vapour can trigger a multitude of symptoms, mainly neuro-psychological disorders, such as tremor or erethism [32,33], which may persist in a reduced intensity even after the end of the exposure [32,34]. Further typical symptoms include a dark discolouration of the gum, gingivitis and renal damage [32,33]. 

In order to curb anthropogenic mercury pollution and to avoid the resulting health and environmental impacts in the future, the U.N. adopted the Minamata Convention on Mercury in 2013 [6]. This convention provided and established regulations to diminish mercury-using practices [6]. Although most of the participating countries have already signed and ratified this convention [35], mercury pollution in the artisanal and small-scale gold mining sector remains a major challenge. The aim of this systematic review was to examine the situation of occupational mercury exposure and mercury-related health effects among ASG miners in middle- and low-income countries in order to give a comprehensive overview on affected workers and their symptoms and diseases.

## 2. Materials and Methods

### 2.1. Conceptualization and Literature Research

The foundation of this systematic review was the “The Preferred Reporting Items for Systematic reviews and Meta-Analyses (PRISMA)” checklist [36]. A systematic study protocol was submitted to and accepted by the International prospective register of systematic reviews (PROSPERO), in order to ensure the methodological quality of this review. The review protocol is available online (PROSPERO-ID: CRD42021235289).

The main research question was developed according to the PECO (Population, Exposure, Comparison, Outcome) criteria [37]: How does a direct ongoing occupational mercury exposure (E) among gold miners (P) influence their health (such as mercury-related symptoms, diseases and intoxication) (O) in comparison to individuals without an occupational mercury exposure (C) (Table 1)?

Exclusively, studies with a peer-reviewed full-text article in English or German were included. These articles had to be published between 1 January 1980 and 31 December 2020. The study design was restricted to the inclusion of prospective studies, observational studies, cross-sectional studies, case-control studies, systematic reviews, non-systematic reviews and meta-analyses. All other kinds of studies were intentionally excluded. Furthermore, only studies that use the amalgam method (or another mercury-related method of gold mining) as well as studies in low- and middle-income countries were included.

To perform a systematic literature research, a search string was created considering three categories: population, exposure and outcome. The selected keywords were combined with the Boolean operators AND (to combine the topics) and OR (to combine the words in a category) to ensure the creation of senseful and complete results. Several words were also ended with *, so that all possible endings were included in the search (for example, symptom* can mean symptom as well as symptoms or symptomatic etc.). For every database (PubMed^®^, EMBASE^®^ and Web of Science™), a specific search string was individually established according to the given guidelines. 

The research was implemented on the 16 February 2021 by one reviewer (Kira Taux). In addition, the complete reference lists of certain key reviews [38,39,40,41,42] were also screened for eligible literature.

### 2.2. Screening Process

All duplicates were removed from the list of results; then the selection of literature started according to the a priori designed review protocol. The entire screening process was conducted by the same two independent reviewers (Kira Taux and Andrea Kaifie).

A screening of the articles’ titles, then abstracts and finally full texts was performed by these reviewers to determine whether the aforementioned inclusion criteria were met or not. After suitable studies were selected by each reviewer, the results were compared, and disagreement was solved by discussion until consensus was achieved. Each exclusion of a study was individually documented, including the specific reason for exclusion. 

### 2.3. Data Extraction (Quantitative Assessment)

A table containing the following categories was created to outline all relevant data: author, year, study design, setting, time, participants, exposure, measurements, outcome, effect parameters. 

Depending on the given statistics, the statistical mean was preferred to report any kind of socio-demographic data (e.g., age or working time), while median values were reported as outcome (for example, laboratory parameters). If no effect parameters were given, odds ratios (OR) were calculated by one author (Andrea Kaifie) if the underlying data were available [43,44,45,46,47,48]. 

All data were extracted and summarized by one reviewer (Kira Taux), while the second reviewer (Andrea Kaifie) controlled the precision and completeness of the data extraction. Although considered, the creation of a meta-analysis out of the selected studies was not possible because the given data were too heterogeneous.

### 2.4. Bias Assessment (Qualitative Assessment)

The methodological quality evaluation of each included study was based on the assessment of the following items: selection bias, performance bias, detection bias, attrition bias, reporting bias and other source(s) of bias. Considering this classification, a table for the assessment was established according to the following references. 

To evaluate the selection bias, two subsidiary categories (bias and confounder) were considered according to questions 14–26 of the checklist first published by Downs and Black [49]. Answer options were yes, no or unable to determine. If all questions were answered with no, this category was rated as low risk; if at least one question was answered with yes, the category was rated as high risk. If at least one question was answered with unable to determine, the category was rated as unable to determine. The checklist had to be adapted to the design of the included studies, which was mainly cross-sectional. Therefore, questions about participants’ blinding and their compliance concerning the intervention (numbers 14 and 19) were considered as not applicable since the studies did not deal with any intervention. The questions concerning the follow-up (numbers 17 and 26) could not be answered adequately because none of the studies had a follow-up. Ultimately, the questions dealing with random sequence generation and allocation concealment (numbers 23 and 24) were omitted since none of the studies was randomized.

The remaining bias categories (performance, detection, attrition, reporting and other source(s) of bias) were evaluated according to the Cochrane Collaboration [50]. Each topic was rated in analogy to the aforementioned categories: the rating was low risk when all questions of an item were judged as low risk, the rating was high risk when at least one question was assessed as high risk and the rating was unclear risk when at least one question was answered as unable to determine. 

A conclusive judgement over all categories was done after all items were evaluated and controlled: studies with an overall low risk of bias had all items assessed as low risk, studies with an overall high risk of bias had at least one category rated as high risk and studies with an unclear risk of bias had at least one category judged as unable to determine.

A blank protocol for this bias assessment is attached as Appendix A.

## 3. Results

### 3.1. Literature Research and Screening Process

The literature research yielded in a total of 10,589 results. After the exclusion of duplicates, 6562 publications were finally taken into consideration. Although the reviewed publications included studies from all over the world, a major portion had to be excluded due to failure to meet the aforementioned inclusion criteria. Title and abstract screening led to the further exclusion of 6464 studies, while the literature review of selected articles [38,39,40,41,42] yielded 5 additional studies to be included in the full-text screening. The subsequent evaluation of the remaining 103 full-texts resulted in the exclusion of 84 articles due to their failure to fulfil the following eligibility criteria: study population (*n* = 55), outcome (*n* = 16), exposure (*n* = 6), study design (*n* = 4), language (*n* = 2) or context (*n* = 1). Eventually, 19 publications were included in this systematic review [43,44,45,46,47,48,51,52,53,54,55,56,57,58,59,60,61,62,63] (Figure 1).

### 3.2. Data Extraction (Quantitative Assessment)

The selected articles included 18 cross-sectional studies [43,44,45,46,47,51,52,53,54,55,56,57,58,59,60,61,62,63] and 1 case series [48], whose publication years ranged from 1993 to 2020 [43,44,45,46,47,48,51,52,53,54,55,56,57,58,59,60,61,62,63]. The following geographical regions were covered:Africa: Ghana [43,44,51,52], Tanzania [45,53], Zimbabwe [46,54], Sudan [55], Burkina Faso [56], Uganda [57].Asia: Indonesia [46,47,58], Pakistan [59,60].South America: Brazil [48,61], Ecuador [62,63] (all Table 2).

#### 3.2.1. Africa

##### Ghana

The first study by Afrifa et al. focused on mercury’s impact on kidney function and discovered that Hg exposure among gold miners was significantly positively associated with elevated levels of urine protein and serum creatinine, while its association with eGFR was significantly negative [43]. None of the surveyed symptoms showed any statistically significant association with mercury exposure [43]. 

Concerning the thyroid function of gold miners, Afrifa and co-authors found a significantly negative association between mercury blood levels with T3 as well as T4 concentrations; in contrast, its association with TSH was non-significantly positive [51]. However, all measured thyroid values were still in a physiological range [51]. 

The connection of mercury with blood pressure was illustrated by the study from Rajaee et al. [44]. A significant association between mercury exposure and hypertension could not be demonstrated for miners versus residents [44].

Mensah et al. observed in their study that mercury exposure did not show significant associations with any medical symptoms for all miners, while the association with numbness was statistically significant for the sub-group with previous work experience in another mine [52].

##### Tanzania

The first study described a variety of symptoms to be more frequent in exposed than in controls, in particular neuro-psychological symptoms, increased salivation or discoloured gums [45]. However, the association between the surveyed symptoms and Hg exposure lacked significance for miners compared to residents [45]. 

The second study detected a variety of symptoms, such as neuro-psychological disorders (e.g., trembling or numbness), gingivitis or several respiratory symptoms in gold miners [53]. The combination of certain observed symptoms with elevated mercury levels in hair led to the diagnosis of mercury intoxication in approximately 10% of the examined miners [53].

##### Zimbabwe

Bose-O’Reilly et al. investigated the medical consequences of occupational mercury exposure in children in Zimbabwe and Indonesia [46]. Certain symptoms, such as neuro-psychological disorders (e.g., ataxia or dysdiadochokinesia), increased salivation or discoloured gums, were more frequent in exposed children [46]. In addition, mercury exposure was significantly associated with certain neuro-psychological disorders, for example, ataxia or dysdiadochokinesia [46]. 

The situation in Zimbabwe was also analysed regarding the disease burden presented in DALYs (Disability-Adjusted Life Years) [54]. The examinations also revealed that a number of symptoms were significantly more common in miners, for example, certain neuro-psychological symptoms [54]. Approximately 3% of the Zimbabwean population was occupationally exposed to Hg, while a total of 2% (which was equivalent to be around 72% of all miners) was considered to be intoxicated, which corresponded to 95,400 DALYs triggered by Hg exposure in the context of ASGM [54].

##### Sudan

Concerning gold miners in Sudan, one study determined variations in thyroid hormones and demonstrated that both TSH and TT4 were significantly elevated in miners in comparison to a control group, while FT3, FT4 and TT3 were significantly reduced [55]. These results were considered generally compatible to the laboratory parameters of hypothyroidism [55].

##### Burkina Faso

Highly exposed workers showed certain medical symptoms, such as neuro-psychological disorders (for example, headache or trembling), thoracic pain or cough [56]. A statistically significant positive association was observed between the mercury values in urine with problems to grab as well as with thoracic pain [56].

##### Uganda

Wanyana et al. described that observed symptoms showed a significant different distribution according to sex: males reported more frequently about headache, while females reported more frequently about psychiatric disorders or memory problems [57]. In addition, mercury exposure was statistically significantly associated with neuro-psychological disorders (such as headache, numbness, dizziness), certain kinds of pain and respiratory symptoms [57].

#### 3.2.2. Asia

##### Indonesia

Following Bose-O’Reilly et al., the examination revealed that certain symptoms, such as certain neuro-psychological disorders, were significantly more frequent in exposed persons [47]. Moreover, cases of mercury intoxication were significantly more frequent in exposed participants [47]. 

Ekawanti and Krisnayanti focused on changes in haematological and renal parameters and highlighted that miners’ haemoglobin and haematocrit was significantly reduced in comparison to a control group, which led to a higher frequency of anaemia [58]. In addition, urine protein was significantly elevated as well, leading to a frequent proteinuria among miners [58].

##### Pakistan

Khan et al. demonstrated that miners complained about various symptoms, such as neuro-psychological disorders, kidney diseases or different kinds of pain [59].

The second study by Riaz et al. also observed that gold miners showed a wide variety of symptoms, for example, gastrointestinal disorders, kidney diseases or respiratory symptoms [60].

#### 3.2.3. South America 

##### Brazil

Lacerda et al. examined the visual performance of gold miners and observed a reduced perimetric area in gold miners. In addition, the colour vision was also significantly worse compared to controls [61]. However, none of the calculated associations or correlations were statistically significant for miners [61].

The study by Branches et al. detected various symptoms in gold miners, for example, neuro-psychological disorders, certain kinds of pain or gingivitis [48]. Nevertheless, the associations between mercury exposure and diagnosed symptoms were not statistically significant [48].

##### Ecuador

The first study presented a significantly positive association between mercury in blood and urine with both tremor and reaction time, while their association with postural sway showed a significantly inverse association [62].

The second study highlighted that a fluctuating proportion of the clinically examined gold miners manifested particular symptoms, such as neuro-psychological disorders, discolouration of the gums or social problems [63]. A statistically significantly positive correlation between the urinary mercury content and the number of medical symptoms could be detected [63].

### 3.3. Risk of Bias Assessment (Qualitative Assessment)

The detailed results of the methodological quality assessment can be found in Table 3. The overall risk of bias rating is attached in the Appendix A. Four studies were evaluated with an overall low risk of bias [43,45,46,47], while fifteen studies were assessed with an overall high risk of bias [44,48,51,52,53,54,55,56,57,58,59,60,61,62,63]. This rating as high-risk was based on a substantial potential for internal validity bias [44,48,52,53,55,56,58,59,60,61,62], internal validity confounder [48,52,53,54,55,56,58,59,60,62], detection bias [44,48,53,54,56,57,58,59,61,63] and/or attrition bias [44,48,51,53,55,56,58,59,60,61]. Other remarks that could not be assigned to the fixed categories but could lead to an increased risk of bias were identified for nine studies [43,48,53,54,57,58,59,60,61].

## 4. Discussion

### 4.1. Data Extraction (Quantitative Assessment)

Although a systematic literature research was conducted, the eligibility criteria only applied to 19 studies, which indicates that there is only limited literature available on the specific topic of mercury-related health effects in ASG miners.

In summary, mercury exposure in the ASGM sector was reported to cause an extraordinarily wide variety of symptoms in diverse organ systems [43,44,45,46,47,48,51,52,53,54,55,56,57,58,59,60,61,62,63]. Interestingly, many symptoms such as pain, hair loss or cough were largely unspecific [43,44,45,46,47,48,51,52,53,54,55,56,57,58,59,60,61,62,63] and not easy to relate to mercury exposure. However, a substantial part of the reported symptoms could be classified to mercury-related neuro-psychological disorders (e.g., tremor, ataxia, memory problems) [43,45,46,47,48,52,53,54,56,57,59,62,63], although the extent varied considerably among the studies. It also has to be considered that working in the ASGM sector in low- or middle-income countries is related to low safety standards and a high probability to suffer from work-related injuries [13]. Challenging working conditions can lead to a high psychological burden, which may cause unspecific somatic symptoms. In a variety of countries with ASGM, medical care availability is often restricted, in particular in terms of occupational health. Therefore, a medical undersupply of work- and non-work-related diseases has to be assumed. Nevertheless, it must be underlined that mercury-related symptoms mean a severe burden of disease in affected persons, in particular for gold miners, who show a high risk of occupational-related mercury exposure.

The organ specific toxicity of mercury exposure has been described in animal studies before [64,65,66]. Akgül and colleagues showed that exposure to mercury vapour caused histological renal damage in rats [64]. Renal damage caused by mercury exposure also has been described in humans, where significantly elevated levels of urine protein or serum creatinine could be observed [43,58]. However, two included studies in this review could not demonstrate significant changes in proteinuria [46,54]. Regarding neuro-psychological abnormalities, Altunkaynak and colleagues observed histological damages in rats’ cerebellum after exposure to mercury vapour [65]. In addition, a further toxicological study in mice detected that post-natal exposure to mercury vapour affects the neuro-behavioral function, such as locomotive activity [66]. These observations support our findings that mercury exposure is in particular connected to neuro-psychological abnormalities [43,45,46,47,48,52,53,54,56,57,59,62,63].

Occupational mercury exposure is not limited to ASGM; it can also occur in other industrial sectors [67,68,69], such as in fluorescent lamp production companies [67]. Mercury-exposed labourers from the fluorescent lamp industry showed statistically significant more frequent psychological symptoms and a worse performance in neuro-psychological tests in comparison to non-exposed controls [67]. These findings confirmed our results, where mercury-exposed gold miners suffered significantly more often from neuro-psychological disorders [45,46,47,54] and showed somewhat significant worse results in neuro-psychological tests [45,46,47,54].

Another sector with an occupational exposure to mercury is the e-waste industry [68]. Decharat et al. examined e-waste workers and highlighted that neuro-psychological disorders, such as headache, were statistically significantly more frequent in exposed workers in comparison to office staff from the same company [68]. These results only partly agree with the findings of this review. For example, we could not observe statistically significant differences for headache between exposed and non-exposed groups [43,45,46]. Only one study could observe a statistically significant association between mercury exposure and headache [57]. However, headache must be considered as an unspecific symptom, which can be caused by a wide range of circumstances and diseases, such as migraine, meningitis or intra-cranial tumours. Consequently, this symptom could have also been caused by other circumstances or pre-existing conditions.

In addition, chlor-alkali workers are occupationally exposed to mercury [69]. These workers showed, according to Neghab et al., a statistically significant association between mercury exposure and neuro-psychological disorders (e.g., memory problems), while other associations with symptoms, such as tremor, showed no statistical significance [69]. However, only two studies in this review found significant differences regarding memory problems [45,47], but in contrast, we detected three studies with non-significant differences [45,46,54]. The reviewed studies used different methods to detect memory problems, from self-reported outcomes to objective neuro-psychological tests [45,46,47,54]. In particular, self-reported memory problems [45,46] are difficult to compare because of a missing generally applicable definition of memory problems, which makes them prone to a recall bias. Similar differing findings have been observed for tremor in miners or mercury-exposed participants [45,46,47,54], where the results differed both in subjective symptoms and clinical examinations [45,46,47,54]. In has to be considered that health consequences of mercury exposure are highly dependent of its chemical form, exposure pathway and dose. In particular the inhalation of inorganic mercury during the amalgamation process leads to a high absorption of and therefore high body burden of toxic metallic mercury. Although ASGM activities are mainly related to occupational inorganic mercury exposure, a burden with organic mercury compounds may be attributed by nutritional habits, such as fish or the ingestion of plant production products. These variables could at least partly explain the differing results.

### 4.2. General Methodology

Since all included studies were cross-sectional studies or case series, none of them was able to randomize their participants into groups [43,44,45,46,47,48,51,52,53,54,55,56,57,58,59,60,61,62,63]. In addition, a follow-up was not available [43,44,45,46,47,48,51,52,53,54,55,56,57,58,59,60,61,62,63]. This would have been helpful in order to understand the long-term effects of mercury and the clinical course in an exposed population.

Considering the study design, the availability of a control group was handled differently among the studies. The majority of the studies had a control group [44,45,46,47,48,53,54,55,56,58,59,61,62]; only six studies lacked a comparable group [43,51,52,57,60,63]. In addition, the composition of control groups was very heterogeneous among the studies. Some control groups consisted exclusively of indirectly exposed participants (e.g., residents from the same area) [48,56,58,59,62] or non-exposed controls (e.g., participants with no known mercury exposure) [54,55]. In contrast, six studies included more than one control group, respectively [44,45,46,47,53,61].

The included studies also differed in terms of data collection and diagnostic procedures [43,44,45,46,47,48,51,52,53,54,55,56,57,58,59,60,61,62,63]. The data were mainly collected through measurements of defined parameters [43,44,45,46,47,48,51,52,53,54,55,56,57,58,59,60,61,62,63], surveys [43,44,45,46,47,48,51,52,53,56,57,58,59,60,61,62] and/or clinical examinations [44,45,46,47,48,53,54,55,56,57,61,62,63]. Since no generally applicable definition of the signs and symptoms of a mercury intoxication existed, the diagnosis standards varied between studies, as well [45,46,47,53,54,63].

### 4.3. Specific Methodology

Six studies reported contradictory data in their manuscripts (see Table 3) [43,48,57,59,60,61]. In addition, one of these studies did not provide all the required information, such as the total number of included participants [59], which made it difficult to assess the results.

Another methodological aspect was the inhomogeneous inclusion of different job categories in the miners’ study groups or in the control group. Exemplarily, three studies also included other occupations, such as gold traders, in the miners’ exposure group [56,57,63]. Therefore, a comparability between the individual studies in general, and in particular their specific results, was limited. Moreover, the study conducted by Ekawanti and Krisnayanti also included child gold miners in the control group [58]. Unfortunately, this point was only mentioned in the Section 4, and no further information was made available in this regard [58].

### 4.4. Strengths and Limitations of This Review

The key strength of this review is the application of a systematic methodology according to the PRISMA scheme [36]. The underlying review protocol was drafted in advance and submitted to PROSPERO (PROSPERO-ID: CRD42021235289) to secure standard methodological claims and transparency. A further strength was the risk of bias assessment, which was created on the basis of relevant literature in order to classify the different findings of the included studies [49,50]. This review provided an overview of the extensive health-related problems caused by ongoing inorganic mercury exposure.

In contrast, the major limitation of this review was the failed attempt to carry out a meta-analysis due to the unsuitable data for this procedure. Another restriction was the limited publication period, ending on 31 December 2020. Therefore, articles that were published later and could have also fulfilled the eligibility criteria could not be considered.

## 5. Conclusions

This systematic review underlines the substantial adverse health effects in ASG miners in low- and middle-income countries. Research should therefore continue to focus on the situation of workers in the ASGM sector. More high-quality studies are urgently necessary, as most of the included studies only had a low methodological quality, resulting in a high risk of bias. These should include a defined control group, a clear definition of mercury-related diseases and a diagnostic standard to detect mercury intoxication. A comprehensive assessment of confounding factors of reported symptoms and diseases is necessary, as well, since a variety of symptoms observed in the included studies were quite unspecific. Due to the cross-sectional design, a causal relation was difficult to derive. Prospective studies that detect a clear causation between mercury exposure and mercury-related outcome are urgently required in order to increase the pressure for change. Mercury remains a significant health threat; this has been described in several toxicological animal studies before and should be underlined by epidemiological analyses in humans in the ASGM sector [64,65,66].

In the past, a first attempt was made to reduce the mercury burden with the Minamata Convention that initiates National Action Plans in the affected countries [6,35]. As the presented results indicate, these efforts were not as sufficient as intended. Mercury still causes serious health problems in ASG miners, and the topic demands intensified attention. Awareness must be raised in miners and their environment. Mercury-related symptoms can persist for a long time, even after termination of the corresponding exposure [32,34]. Hence, governments should take action to improve the working conditions for miners. Miners need to be convinced of replacing mercury-containing practices in gold extraction with non-hazardous techniques, as have already been attempted in model projects [70,71]. Finally, mercury-intoxicated miners must also be given the opportunity to receive sufficient medical care.

## Figures and Tables

**Figure 1 ijerph-19-02081-f001:**
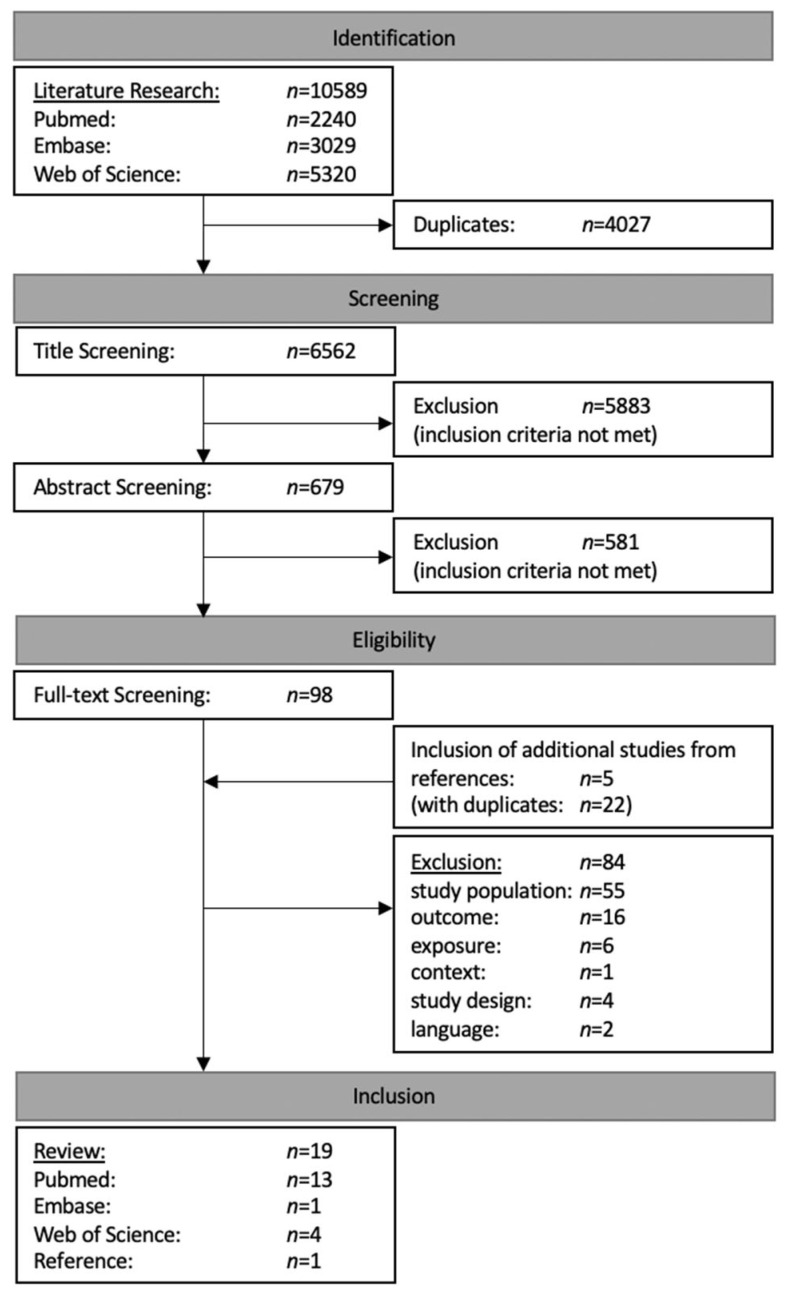
Flow chart of literature research and screening process.

**Table 1 ijerph-19-02081-t001:** Inclusion and Exclusion Criteria.

PECO Scheme	Inclusion Criteria	Exclusion Criteria
Population	Workers (adults as well as children and adolescents under the age of 18 years) with an ongoing occupation as gold miner in middle- and low-income countries	Adults as well as children and adolescents under the age of 18 years with no activity in gold mining; gold miners who interrupted their gold mining activities; former gold miners; residents; workers from high income countries
Exposure	Ongoing direct occupational-related mercury exposure; use of the amalgamation method for gold extraction	No ongoing direct occupational-related mercury exposure; use of other methods for gold extraction (e.g., cyanide)
Comparison (if available)	Adults as well as children and adolescents under the age of 18 years with no direct gold mining activities; residents	Workers (children and adolescents under the age of 18 years and adults) with direct relation to gold mining activities; former gold miners
Outcome	Primary outcome: all health outcomes must be a direct consequence of mercury exposure; mercury-related diseases; symptoms of acute and chronic mercury intoxication; long-time health	Studies that do not match the inclusion criteria

**Table 2 ijerph-19-02081-t002:** Data extraction.

Author, Year	Study Design, Setting, Time, Participants, Exposure	Measurements, Examinations	Outcome	Effect Parameters (Bold Indicates Statistically Significance)
Afrifa et al. (2017) [43]	cross-sectional study Bibiani/Western Region, Ghana year not reported gold miners (occupational (direct) Hg exposure): n = 110 (male, various inclusion criteria, e.g., ≥1 year occupational Hg exposure, no kidney-affecting disease) exposed (B–Hg ≥ 5.0 µg/L): n = 61 (mean age 35.8 years, mean work duration 14.7 years) non-exposed (B–Hg < 5.0 µg/L): n = 49 (mean age 34 years, mean work duration 10.8 years)	laboratory: spot urine sample: proteinuria blood sample: Hg, serum creatinine, eGFR examination: questionnaire: socio-demographics, anamnesis, occupation occupational assessment interview	laboratory (only exposed): significantly elevated levels: B–Hg (mean 18.4 µg/L) urine protein (mean 41.7 mg/dL) serum creatinine (mean 2.2 µmol/L) significantly reduced levels: eGFR (mean 57 mL/min/1.73 m^2^) examination: symptoms associated with Hg exposure: non-significant more frequent in exposed: skin rash, cough, fever, itchy eyes, fatigue, headache, muscle ache, numbness, hair loss non-significant less frequent in exposed: metallic taste	correlation: B–Hg: significantly positive: proteinuria **(r = 0.7)** significantly negative: eGFR **(r = −0.8)** odds ratio OR (95% CI): Hg exposure (age-adjusted): urine protein ≥ 10 mg/dL: OR = **50.3 (11–230.5)** serum creatinine > 106 µmol/L: OR = **101.1 (25.2–404.9)** eGFR ≤ 90 mL/min/1.73 m^2^: OR = **263.2 (48.8–1420)** Hg exposure and symptoms (OR (95% CI)) (calculated by the authors of this systematic review from available data in the original publication): neuro-psychological symptoms: fatigue (OR = 1.1 (0.5–2.6)), headache (1.3 (0.5–3)), numbness (OR = 1.9 (0.9–4.1)) other symptoms: skin rash (OR = 1.4 (0.7–3)), cough (OR = 1.7 (0.8–3.6)), fever (OR = 1.1 (0.5–2.4)), itchy eyes (OR = 1.2 (0.5–2.9)), muscle ache (OR = 1.5 (0.7–3.2)), hair loss (OR = 1.2 (0.2–7.6)), metallic taste (OR = 0.7 (0.3–1.5))
Afrifa et al. (2018) [51]	cross-sectional study Bibiani/Western Region, Ghana January 2017–March 2018 gold miners (occupational (direct) Hg exposure): n = 137 (male, various inclusion criteria, e.g., ≥1 year occupational Hg exposure, no diseases affecting liver or thyroid) exposed (B–Hg ≥ 5.0 µg/L): n = 80 (median age 26 years, median work duration 8 years) non-exposed (B–Hg < 5.0 µg/L): n = 57 (median age 30 years, median work duration 4 years)	laboratory: blood sample: thyroid hormones: T4, T3, TSH Hg questionnaire: socio-demographics, anamnesis, occupation	laboratory: exposed: significantly elevated levels: B–Hg (median 8 µg/L) significantly reduced levels: − T4 (mean 5.4 µg/dl) − T3 (mean 1.5 nmol/L) non-significantly elevated levels: TSH (mean 1.7 mIU/L) both groups: normal range for thyroid parameters	correlation: B–Hg: significantly negative: T4 **(r = −0.7)**, T3 **(r = −0.3)** non-significantly positive: TSH (r = 0.1)
Rajaee et al. (2015) [44]	cross-sectional study Upper East Region, Ghana Kejetia (miners + residents) Gorogo (controls) May–July 2011 gold miners (occupational (direct) Hg exposure): n = 70 (n = 42 male, n = 28 female, mean age 30.6 years) residents (environmental (indirect) Hg exposure): n = 26 (n = 7 male, n = 19 female, mean age 33.8 years) controls (no known (background) Hg exposure): n = 75 (n = 34 male, n = 41 female, mean age 51.5 years)	laboratory: spot urine sample (n = 91): Hg hair sample (n = 69): MeHg examination: interview: socio-demographics, anamnesis, occupation, lifestyle medical parameters: pulse blood pressure: SBP, DBP, MAP, PP	laboratory (only miners’ median values reported): U–Hg: significantly elevated levels in miners (4.2 µg/L, 5.2 µg/L (SG-adjusted)) (trend: miners > residents > controls) H–Hg: significantly elevated levels in miners (0.9 µg/g) (trend: miners > residents > controls) examination (only miners): mean blood pressure: 122.6/75.2 mmHg hypertension: n = 11	correlations (only Kejetia): significantly negative: U–Hg (SG-adjusted) + pulse **(****ρ****= −0.2)** associations (only miners): H–Hg: non-significantly positive: PP (ß = 1.9), SBP (ß = 1.4), MAP (ß = 0.1) non-significantly negative: DBP (ß = −0.5), pulse (ß = −0.3) U–Hg (SG-adjusted): non-significantly positive: PP (ß = 0.2), SBP (ß = 0.1) non-significantly negative: DBP (ß = −0.1), pulse (ß = −0.02), MAP (ß = −0.04) odds ratio (OR (95% CI)) (calculated by the authors of this systematic review from available data in the original publication) (Hg exposure, miners vs. residents): hypertension: (OR = 0.8 (0.3–2))
Mensah et al. (2016) [52]	cross-sectional study Prestea/Western Region, Ghana 2012 gold miners (occupational (direct) Hg exposure): n = 343 (n = 323 male, n = 20 female, age 15–70 years, mean age 29.5 years, work duration 1–38 years, mean work duration 7.2 years) exposure: − exposed (U–Hg ≥ 5.0 µg/L): n = 160 − non-exposed (U–Hg < 5.0 µg/L): n = 183 workplace (previous occupation in another mine): n = 25	laboratory: morning urine sample: Hg examination: questionnaire: socio-demographics, anamnesis, occupation interview + observation: occupational assessment	laboratory: exposed: U–Hg 5–50.5 µg/L (mean 14.8 µg/L)	associations (Hg exposure): non-significant in all miners (n = 343): neuro-psychological symptoms: headache (χ^2^ = 0.7), fatigue (χ^2^ = 0.01), insomnia (χ^2^ = 1), numbness (χ^2^ = 0.00) other symptoms: red eyes (χ^2^ = 0.3), skin rash (χ^2^ = 3.5), cough (χ^2^ = 0.2), fever (χ^2^ = 0.1), metallic taste (χ^2^ = 0.4), muscle ache (χ^2^ = 0.1), sinusitis (χ^2^ = 0.4), hair loss (χ^2^ = 0.04) significant in miners with a previous occupation in another mine (n = 25): numbness **(χ^2^ = 5)**
Bose-O’Reilly et al. (2010a) [45]	cross-sectional study Tanzania: Rwamagasa/Geita District Katoro October–November 2003 gold miners (occupational (direct) Hg exposure): n = 138 (from Rwamagasa) non-smelters: n = 34 (n = 20 male, n = 14 female, age 14–50 years, mean age 26.6 years) smelters: n = 104 (n = 87 male, n = 17 female, age 14–57 years, mean age 33.8 years) residents (environmental (indirect) Hg exposure): n = 52 (from Rwamagasa, n = 21 male, n = 31 female, age 11–57 years, mean age 32.3 years) controls (no known (background) Hg exposure): n = 31 (from Katoro, n = 12 male, n = 19 female, age 15–51 years, mean age 32.4 years) → exposed: smelters, non-smelters, residents	laboratory: blood sample: Hg urine sample (n = 218): Hg hair sample (n = 188): T–Hg, I–Hg, O–Hg examination: questionnaire: anamnesis, Hg exposure clinical examination with special focus on neurology neuro-psychological tests: memory test, matchbox test, pencil tapping test, Frostig score diagnosis of intoxication: algorithm including symptoms + laboratory parameters	laboratory (only miners’ median values): significantly elevated levels in exposed groups: smelters >> non-smelters > residents > controls U–Hg: non-smelters 1 µg/L (0.8 µg/gCr), smelters 5.9 µg/L (3.6 µg/gCr) B–Hg: non-smelters 1.6 µg/L, smelters 2.5 µg/L hair: − T–Hg: non-smelters 0.5 µg/g, smelters 0.8 µg/g − I–Hg: non-smelters 0.1 µg/g, smelters 0.4 µg/g examination: significantly more frequent in exposed worsened health neuro-psychological symptoms: memory problems, tiredness, tremor finger-to-nose + eyelid, sadness, problems to find words, ataxia of gait, sensory disturbance, abnormal ASR + BSR other symptoms: less appetite, salivation, discoloured gum no significant difference: neuro-psychological symptoms: sleep problems, concentration problems, thinking problems, nervousness, headache, numbness, problems with impetus, dysdiadochokinesia, intentional tremor heel-to-shin, ataxia heel-to-shin, bradykinesia, hypomimia, abnormal PSR other symptoms: loss of hair, metallic taste, less muscle strength, weakness, eyestrain problems, palpitations, nausea, stomatitis, gingivitis neuro-psychological tests: significantly worse results in exposed: pencil tapping test, matchbox test no significant differences: memory test, Frostig score intoxication: smelters (n = 104) n = 25, residents (n = 31) n = 1	odds ratio (OR (95% CI)) (calculated by the authors of this systematic review from available data in the original publication) (Hg exposure, gold miners vs. residents): neuro-psychological symptoms: sleep problems (OR = 1.2 (0.6–2.5)), tremor (OR = 3.1 (0.9–10.9)), nervousness (OR = 1.2 (0.5–2.9)), sadness (OR = 1.9 (0.9–3.7)), headache (OR = 1.3 (0.7–2.4)), numbness (OR = 1.1 (0.6–2.1)) other symptoms: loss of appetite (OR = 1.7 (0.8–3.6)), hair loss (OR = 2.7 (0.1–53.4)), metallic taste (OR = 1.1 (0.2–5.8)), salivation (OR = 2.5 (1–6.4)), palpitations (OR = 1.4 (0.7–2.7)), nausea (OR = 1.1 (0.5–2.4))
Harada et al. (1999) [53]	cross-sectional study Tanzania: seven gold mines near Lake Victoria one gold mine in the inland three fishing villages around Lake Victoria one fishing village in the inland Mwanza/State of Mwanza 1996–97 (three spot investigations) gold miners (occupational (direct) Hg exposure): n = 150 (age 7–70 years, n = 136 male, n = 14 female, work duration 3–10 years) fishermen (environmental (indirect) Hg exposure): n = 103 (fishermen + families, age 6–70 years, n = 87 male, n = 16 female) controls (no known (background) Hg exposure): n = 19 (inhabitants of Mwanza, age 0.5-46 years, n = 11 male, n = 8 female)	laboratory: hair: T–Hg, MeHg (n = 9) fish (Lake Victoria): T–Hg examination (n = 225): questionnaire clinical examination with focus on neurology diagnosis of intoxication: elevated T–Hg + symptoms	laboratory: gold miners: mean T–Hg 1–81.9 ppm mean T–Hg 1–4 ppm (exclusion of six cases >50 ppm) MeHg/T–Hg ratio (n = 3 from each group, marker for direct non-dietary Hg exposure): higher in gold miners (1–20.5%) fish: tilapia 63 ppb, sardine 12 ppb, catfish 8.9 ppb examination of gold miners (n = 118): neuro-psychological symptoms (trembling 21.2%, headache 11.9%, numbness 9.3%, sensory disturbance 8.5%, taste problems 8.5%, tremor 8.5%, abnormal reflex 5.1%/2.5%, memory problems 7.6%, smell problems 5.9%, insomnia 5.1%, vertigo/dizziness 5.1%, neurasthenia 3.4%, night blindness 3.4%) pain (chest pain 8.5%, limb pain 5.9%) respiratory problems (dyspnoea 7.6%, cough 6.8%) other symptoms: palpitation (5.9%), gingivitis (13.6%) mild I–Hg intoxication: n = 14	–
Bose-O’Reilly et al. (2008) [46]	cross-sectional study Sulawesi/Indonesia, Kadoma/Zimbabwe 2003–04 gold miners (occupational (direct) Hg exposure): n = 80 (median age 12 years, n = 60 male, n = 20 female) residents (environmental (indirect) Hg exposure): n = 36 (living in gold mining areas, median age 11 years, n = 11 male, n = 25 female) controls (no known (background) Hg exposure): n = 50 (median age 12 years, n = 24 male, n = 26 female, living in areas without gold mining)	laboratory: spot urine sample: Hg blood sample: Hg hair sample (n = 150): T–Hg, I–Hg, O–Hg examination: questionnaire: anamnesis, Hg exposure, confounders clinical examination with special focus on neurology neuro-psychological tests: memory test, matchbox test, pencil tapping test, Frostig score diagnosis of intoxication: algorithm including symptoms + laboratory parameters	laboratory (only miners’ values): significantly elevated levels in miners + residents (highest burden in miners): B–Hg: median 7.8 µg/L U–Hg: median 10.1 µg/L, 7.1 µg/gCr hair: − T–Hg: median 2.3 µg/g − I–Hg: median 0.9 µg/g elevated in miners > residents: factor U–Hg/B–Hg (U–Hg > B–Hg, miners: 3.8) examination: significantly more common in miners > residents neuro-psychological symptoms (ataxia, dysdiadochokinesia, abnormal reflexes) other symptoms: salivation, metallic taste, blue coloration of gums no significant difference: neuro-psychological symptoms (headache, memory problems, nausea, numbness/prickling feet, concentration problems, sleeping problems, tremor, hypomimia, mento-labial reflex) other symptoms: gingivitis, stomatitis, proteinuria neuro-psychological tests: significantly worse performance in miners + residents: matchbox test, pencil tapping test no significant difference: memory test, Frostig score intoxication: miners: n = 20, residents: n = 4	odds ratio (OR (95% CI)) (calculated by the authors of this systematic review from available data in the original publication) (Hg exposure, gold miners vs. controls): neuro-psychological symptoms: headache (OR = 0.7 (0.3–1.9)), memory problems (OR = 0.9 (0.2–5.8)), numbness/prickling feet (OR = 2 (0.4–10)), sleeping problems (OR = 1.9 (0.2–18.9)), concentration problems (OR = 4 (0.5–34)), dysdiadochokinesia **(OR = 4.6 (1.5–14.4))**, hypomimia (OR = 3.2 (0.2–68.4)) ataxia: heel-to-shin **(OR = 3.3 (1.1–10.5))**, gait **(OR = 5.2 (1.5–18.6))** abnormal reflex: mento-labial **(OR = 0.6 (0.3–1.6))**, ankle jerk **(OR = 5.5 (1.8–17))**, biceps brachii **(OR = 8 (1.8–35.9))**, quadriceps **(OR = 10.4 (1.3–81.7))** tremor (OR = 2 (0.2–19.3)): finger-to-nose (OR = 4.6 (0.2–90.2)), heel-to-shin (OR = 1.9 (0.1–47.7)), eyelid (OR = 1.5 (0.7–3)) other symptoms: salivation (OR = 15 (0.9–262.7)), metallic taste (OR = 8.8 (0.5–159.9)), nausea (OR = 0.9 (0.2–5.8)), blue coloration of gums (OR = 8.8 (0.5–159.9)), gingivitis (OR = 0.6 (0–32.1)), stomatitis (OR = 0.6 (0–32.1)), proteinuria (OR = 0.1 (0–1.7))
Steckling et al. (2014) [54]	cross-sectional study Zimbabwe Kadoma (gold miners) Chikwaka (controls) 2004, (2006) gold miners (occupational (direct) Hg exposure): n = 181 (age 9–75 years, mean age 27 years, n = 122 male, n = 59 female, n = 33 non-smelters, n = 148 smelters, occupational Hg exposure 1–23 years, mean occupational Hg exposure 4 years) controls (no known (background) Hg exposure): n = 91 (age 11–59 years, mean age 24 years, n = 24 male, n = 67 female)	definition of chronic Hg intoxication: algorithm including symptoms + laboratory parameters diagnosis of chronic Hg intoxication disease prevalence in Zimbabwe: socio-demographic data DALY = YLD (years lived with disability) + YLL (years of life lost)	laboratory (miners’ median values): higher values in miners 80–90% miners + few controls exceed limit values U–Hg: 26.1 µg/L, 25.8 µg/gCr B–Hg (n = 152): 11.4 µg/L H–Hg (n = 158): 3.3 µg/L symptoms (miners): medical score sum: significant higher significantly more frequent: neuro-psychological symptoms: tremor at work + finger-to-nose, ataxia of gait, dysdiadochokinesia other symptoms: metallic taste, blue discoloured gum worse results in neuro-psychological tests: Frostig test, pencil tapping test non-significantly more frequent: worsened health neuro-psychological symptoms: heel-to-knee ataxia, heel-to-knee-tremor, abnormal reflex, sleeping problems other symptoms: salivation, proteinuria worse results in neuro-psychological tests: memory test, matchbox test intoxication: n = 131 miners gold miners: total 350,000 adults (≥15 years): 85% (male 70%, female 30%) children (9–14 years): 15% (male 73%, female 27%) chronic Hg intoxication: population: 2% (3% occupational Hg exposure) miners: 72% (adults 72% (male 90%, female 40%), children 76%) 95,400 DALYs (8 DALYs/1000 persons) most affected: male (78,400 DALYs, 13 DALYs/1000 persons) female (17,000 DALYs, 3 DALYs/1000 persons)	–
Tayrab (2017) [55]	cross-sectional study Sudan: Abuhamad gold mining area/River Nile State (gold miners) Khartoum State (controls) August 2012–November 2014 gold miners (occupational (direct) Hg exposure): n = 83 (male, age 18–55 years, mean age 30.5 years, occupation > 6 months: 44.6% wells, 32.5% mills, 16.9% washing, 6% moulding) controls (no known (background) Hg exposure): n = 50 (male, mean age 28.1 years)	laboratory: blood sample: TSH, TT3, TT4, FT3, FT4 clinical examination	significantly elevated levels in miners: TSH (mean 5.1 µIU/mL) TT4 (mean 86.3 pmol/L) significantly reduced levels in miners: TT3 (mean 1.2 ng/dL) FT3 (mean 1.3 pg/mL) FT4 (mean 6.4 ng/dL) elevated in miners: FT4/FT3 ratio (mean 4.8) laboratory results (TSH, FT3, TT3) compatible with hypothyroidism	–
Tomicic et al. (2011) [56]	cross-sectional study Burkina Faso (eight gold mining areas in six regions) Bagassi Bouda Fandjora II + III Mossobadougou Pousghin (Macara) Safané Zinigma year not reported participants: n = 1090 gold miners (occupational (direct) Hg exposure): n = 779 − most susceptible to Hg: n = 93 (n = 82 male, n = 11 female, age 17–56 years, mean age 31.7 years, occupational Hg exposure 1–12 years, mean occupational Hg exposure 4 years) − gold dealers: n = 146 (susceptible: n = 52) − ore washers: n = 151 (susceptible: n = 33) − others (susceptible: n = 8) non-miners (environmental (indirect) Hg exposure): e.g., miners’ families	laboratory: spot urine sample: Hg (n = 93), albumin, creatinine examination: questionnaire: socio-demographics, anamnesis, occupation, Hg exposure, consumption medical parameters	laboratory: U–Hg: 3–3493 µg/L, 4.3–1707 µg/gCr 69% >35 µg/gCr, 16% >350 µg/gCr (reference < 3 µg/gCr) statistically significant trend: dealers > ore washers > others albuminuria: 40.9% (most susceptible miners) examination: more frequent in gold miners: walking problems (8.2%) rhinitis (9.5%) more frequent in susceptible miners: neuro-psychological symptoms (headache 53.3%, sleep problems 25.3%, dizziness 53.8%, tiredness 33%, trembling 31.9%, sensory problems on hands/feet 23.1%, visual problems 30.8%) other symptoms: mouth irritations/wounds (22%), cough (27.5%), thoracic pain (34.1%)	association: significantly positive: ore washing + trembling: **χ^2^ 6.6** packaging Hg + rhinitis: **χ^2^ 5.6** U–Hg: − thoracic pain: **χ^2^ 10.4** − grabbing problems: **χ^2^ 6.8** non-significantly positive: gold dealer + trembling: χ^2^ 3.7 heating Hg + thoracic pain: χ^2^ 3.5
Wanyana et al. (2020) [57]	cross-sectional study Uganda (two mining areas each): Amudat/Karamoja Region Busia/Eastern Region Ibanda/Western Region Mubende/Central Region June–July 2018 gold miners (occupational (direct) and environmental (indirect) Hg exposure): n = 183 (n = 133 male, n = 50 female, age 15–65 years, occupation ≥ ½ year, mean Hg exposure 5.4 years), including extractors processors burners/buyers	laboratory: blood sample (n = 31): Hg urine sample (n = 31): Hg environmental samples (n = 26, water + topsoil): Hg examination: questionnaire: socio-demographics, Hg exposure clinical examination: anamnesis, neurological examination	laboratory: blood: B–Hg 26.3–205 µg/L (median 67.5 µg/L, all samples exceed HBM-II limit value) significantly higher: Mubende, OHS knowledge urine: U–Hg 37.5–296 µg/L (median 70.8 µg/L, all samples exceed HBM-II limit value) significantly higher: Mubende, female, panners, OHS knowledge environment: drinking water: mean 23.8 µg/L (in all samples above WHO limit (6 µg/L)) soil: mean 0.2 µg/L symptoms (examination): least in Ibanda statistically significant more frequent: − female: swollen legs, psychiatric problems, stomachache, memory problems, diarrhoea, respiratory problems − male: headache no significant differences between male + female: injuries, numbness, shaking hands, pain (back, chest, joint, feet), eye problems, general malaise, dizziness, fatigue + stress	odds ratio (OR (95% CI)) (Hg exposure): odds ratio (adjustment for confounders)/odds ratio neuro-psychological symptoms: shaking of hands + head: **(OR = 24.1 (1.7–338.7)/OR = 7.8 (2.7–22))** eye problems: **(OR = 11 (2–62.5)/OR = 9.2 (3.7–23.2))** numbness: **(OR = 8.5 (2.1–34.4)/OR = 7.9 (7.9–3.6))** fatigue + stress: **(OR = 5.4 (1.9–14.9)/OR = 6.5 (3.5–12.3))** headache: **(OR = 4.7 (1.9–11.3)/OR = 6.4 (3.4–12))** dizziness: **(OR = 3.8 (1.5–9.7)/OR = 6 (2.9–12.2))** pain: chest: **(OR = 9 (3.3–24.6)/OR = 8 (4–16))** back: **(OR = 6.2 (2.2–17.5)/OR = 6.1 (3.3–11.2))** joint: **(OR = 3.2 (1.3–8.3)/OR = 6.1 (2.9–12.9))** respiratory problems: **(OR = 3.2 (1–10.1)/OR = 6.8 (2.7–17.4))**
Bose-O’Reilly et al. (2010b) [47]	cross-sectional study Indonesia: Tatelu in North Sulawesi Kerang Pangi in Central Kalimantan August–September 2003 gold miners (occupational (direct) Hg exposure): non-smelters: n = 47 (n = 31 female, n = 16 male, age 19–59 years, mean age 33.7 years for Kalimantan/36 years for Sulawesi) smelters: n = 129 (n = 28 female, n = 101 male, age 19–58 years, mean age 31.9 years for Kalimantan/35.2 years for Sulawesi) residents (environmental (indirect) Hg exposure): n = 84 (n = 78 female, n = 6 male, age 19–57, mean age 31.6 years for Kalimantan/39.1 years for Sulawesi) controls (no known (background) Hg exposure): n = 21 (only from North Sulawesi, n = 4 female, n = 17 male, age 21–46 years, mean age 27.1 years) → exposed: smelters, non-smelters, residents	laboratory: blood sample: Hg urine sample: Hg hair sample: T–Hg, I–Hg, O–Hg examination: questionnaire: anamnesis, Hg exposure clinical examination with special focus on neurology neuro-psychological tests: memory test, matchbox test, pencil tapping test, Frostig score diagnosis of intoxication: algorithm including symptoms + laboratory parameters	laboratory (only miners’ median values): all parameters significantly elevated in exposed (trend: smelter > non-smelter > resident > control): U–Hg: non-smelter 5.3–7.8 µg/L (3.3–3.7 µg/gCr), smelter 10.2–22.4 µg/L (5.3–10.2 µg/gCr) B–Hg: non-smelter 9.2–9.5 µg/L, smelter 10.6–13.3 µg/L hair: − T–Hg: non-smelter 3–3.8 µg/g, smelter 3.9–4.9 µg/g − I–Hg: non-smelter 1.1–1.2 µg/g, smelter 1.3–2 µg/g examination: more symptoms in exposed significantly more frequent in exposed groups: neuro-psychological symptoms (ataxia of gait + heel-to-shin, eyelid tremor, hypomimia) more frequent in different locations: Central Kalimantan: hair loss, salivation, numbness, fatigue North Sulawesi: sleeping problems, fatigue (for smelters) no significant difference: finger-to-nose tremor, dysdiadochokinesia neuro-psychological tests: worse results in all tests for exposed significantly worse results: memory test, pencil tapping test, matchbox test non-significantly worse results: Frostig score intoxication: significantly more cases in exposed (all from Kalimantan, smelters from Sulawesi) smelters: n = 33 (Sulawesi), n = 43 (Kalimantan) non-smelters: n = 4 (Sulawesi), n = 13 (Kalimantan) residents: n = 3 (Sulawesi), n = 21 (Kalimantan)	odds ratio (OR (95% CI)) (calculated by the authors of this systematic review from available data in the original publication) (Hg exposure): intoxication (gold miners vs. residents): Kalimantan: **(OR = 2.8 (1.5–5.4))** Sulawesi: **(OR = 4.6 (1.2–17.3))**
Ekawanti and Krisnayanti (2015) [58]	cross-sectional study Sekotong ASGM/West Nusa Tenggara Province, Indonesia year not reported gold miners (occupational (direct) Hg exposure): n = 71 (male, occupational Hg exposure for ≥5 years, detailed work duration not reported, smokers) non-miners (environmental (indirect) Hg exposure): n = 29 (n = 25 female, n = 4 male, miners’ families as wives + children, living in the mining area for ≥1 year)	laboratory: blood sample: haemoglobin, haematocrit urine sample: Hg, proteinuria hair sample: Hg questionnaire: Hg exposure, duration + handling	laboratory: urine: significantly elevated levels in miners: − urine protein (mean 1.7 g/L) − U–Hg (mean 69.4 µg/L) more frequent in miners: proteinuria (92.6%) blood: significantly reduced in miners: − haemoglobin (mean 12.7 g/dL) − haematocrit (mean 38.2%) more frequent in miners: anaemia (57.7% for standard limits, 67.6% for smokers’ limits) H–Hg: non-significantly elevated in miners: (mean 2.8 µg/L)	–
**Khan et al. (2012) [59]**	cross-sectional study Gilgit-Baltistan Province, Pakistan Astor Dainyor Gulmiti Gupis Haramosh Ishkumen Jaglot Jalalabad Jutal Khari Shimshal year not reported gold miners (occupational (direct) Hg exposure): n = unknown adults: n = unknown (age 18–50 years, male + female) children: n = unknown (age 8–15 years, male + female) control group (environmental (indirect) Hg exposure): n = unknown (male, female, children)	laboratory: blood sample: RBC plasma urine sample: Hg questionnaire: socio-demographics, anamnesis, Hg exposure, consumption significance level: <0.01	laboratory (only mean values for miners): urine: significantly elevated levels of U–Hg adults: male 57.1 µg/L, female 68.5 µg/L (98% exceed WHO limit (50 µg/L)) children: male 24.5 µg/L, female 13.6 µg/L blood (no detailed numbers reported): significantly elevated levels of T–Hg, I–Hg examination (only miners): higher percentage of symptoms symptoms (male, female): kidney disease (56%, 30%), skin rash (38%, 48%), neuro-psychological symptoms (tiredness + headache 67%, 23%; cognitive problems 45%, 20%; sensory problems 31%, 28%; tremor 16%, 9%; memory problems 12%, 7%; abnormal reflexes 6%, 4%; smell + taste problems 16%, 12%; night blindness 9%, 2%; neurasthenia 7%, 8%), pain (chest 53%, 67%; limb 10%, 8%), children: slow growth (67%, 71%), cough (31%, 22%), palpitation (21%, 28%)	–
Riaz et al. (2016) [60]	cross-sectional study Gilgit-Baltistan Province, Pakistan Chalt Chirmish Goharabad Jalalabad Khari Minor Khari Nomal Yashokaldas year not reported gold miners (occupational (direct) Hg exposure): n = 45 (male + female, detailed distribution of age, sex + work duration not reported)	laboratory: blood sample: plasma: T–Hg, I–Hg, O–Hg RBC: T–Hg, I–Hg, O–Hg urine sample: T–Hg, I–Hg, O–Hg hair sample: Hg nail sample: Hg questionnaire: socio-demographics, anamnesis, occupation, consumption significance level: ≤0.01	laboratory (only mean values for urine reported): RBC: results exceed limits of WHO + USEPA plasma: results exceed limits of USEPA (5.8 µg/L) urine: results exceed limits of WHO (50 ng/mL) T–Hg: male 61.4 µg/L, female 51.7 µg/L I–Hg: male 40.5 µg/L, female 36.5 µg/L H–Hg: male 2.7 g/kg, female 1.8 µg/kg N–Hg: male 2.2 µg/kg, female 2.8 µg/kg examination: more health problems in occupational context health problems (male, female): kidney disease (67%, 54%), gastrointestinal problems (stomach problems 89%, 75%; teeth problem 58%, 46%; belly pain 68%, 46%), neck pain (45%, 58%), joints problem (35%, 22%), hernia (25%, 18%), skin burn (75%, 98%), stunted growth in children (56%, 34%), heart problem (45%, 35%), inhalation problem (76%, 86%)	–
**Lacerda et al. (2020) [61]**	cross-sectional study Pará State, Brazil Itaituba (riverines) Serra Pelada (gold miners) year not reported gold miners (occupational (direct) Hg exposure): n = 34 (male, mean age 45.9 years) riverines (environmental (indirect) Hg exposure): n = 10 (male, mean age 40.9 years) controls (only for hue ordering test, no known (background) Hg exposure): n = 41 (male, age-matched, urban inhabitants, no disease affecting the visual outcome, visual acuity 20/20)	laboratory: hair sample: Hg examination: questionnaire: socio-demographics, anamnesis, occupation, consumption, smoking visual test: participants with visual acuity ≥20/40 on both eyes (better one tested) visual perimetry hue ordering test	laboratory: significantly reduced levels in miners: H–Hg examination: perimetry: riverines: all below reference, significantly smaller perimetric area than miners miners: 61.8% below reference hue ordering test: miners: significantly more errors than controls, no significant difference to riverines	associations (only miners): H–Hg: visual perimetry: partial r = 0.1 colour vision: partial r = 0.5 correlations (only miners): non-significant for visual outcome: r = −0.2
Branches et al. (1993) [48]	case series Brazil Tapajós/Pará region Tapajós/Amazonas region 1986–91 participants: n = 55 (hospital patients with suspicion or anamnesis of Hg exposure, age 8–75 years) exposed group (occupational (direct) Hg exposure): − gold shop workers (n = 11, male, mean age 37 years, mean occupational Hg exposure 5.3 years) − gold miners (n = 22, male, mean age 43 years, mean occupational Hg exposure 16.3 years) residents (environmental (non-occupational) Hg exposure): n = 22 (urban inhabitants, n = 11 male, n = 10 female, mean age 37 years)	laboratory: blood sample: Hg spot urine sample: Hg examination questionnaire: anamnesis, occupation clinical examination	laboratory (gold miners): B–Hg: mean 2.2 µg/dL, median 2.3 µg/dL U–Hg (n = 6): mean 35.4 µg/L, median 25 µg/L examination (gold miners): neuro-psychological symptoms (dizziness 45%, headache 45%, tremor 45%/9%, insomnia 41%, numbness 41%, visual problems 41%/18%, memory problems 32%, nervousness 23%, balance problems 18%/14%, coordination problems 18%, fatigue 18%, seagull sign 18%, cramps 14%, fear 14%, ataxia 5%, deep sensibility problems 5%, tactile problems 5%) pain (chest pain 27%, abdominal pain 14%) other symptoms: palpitations (36%), dyspnoea (27%), oedema (27%/18%), hair loss (27%/5%), hepatomegaly (27%) + splenomegaly (23%) (positive anamnesis for malaria), loss of appetite (27%), impotence (23%), weakness (23%), weight loss (23%), tonsillar hypotrophy (18%), no physical pathologies (14%), premature aging (14%), pruritus (5%), gingivitis (5%)	odds ratio (OR (95% CI) (calculated by the authors of this systematic review from available data in the original publication) (Hg exposure): neuro-psychological symptoms: dizziness (OR = 0.8 (0.3–2.7)), headache (OR = 1.2 (0.5–2.9)), tremor (OR = 1.8 (0.5–6.1)), numbness (OR = 1.5 (0.5–4.7)), insomnia (OR = 1.9 (0.5–6.6)), nervousness (OR = 0.3 (0.1–1.1)), visual problems (OR = 4.4 (1–19.4)), memory problems (OR = 2.1 (0.5–8.6)), cramps (OR = 0.7 (0.1–3.6)), fatigue (OR = 1.4 (0.3–7.2)), fear (OR = 0.7 (0.1–3.6)) pain: chest (OR = 3.6 (0.7–21.2)), abdomen (OR = 1 (0.2–5.6)) other symptoms: palpitations (OR = 0.5 (0.1–1.6)), dyspnoea (OR = 1 (0.3–3.8)), loss of appetite (OR = 1 (0.3–3.8)), weakness (OR = 0.5 (0.1–1.9)), hair loss (OR = 2.4 (0.5–11.1)), pruritus (OR = 0.5 (0–5.7)), impotence (OR = 14.1 (0.7–273.4)), weight loss (OR = 2.9 (0.5–17.1)), oedema (OR = 7.9 (0.9–72.1))
Harari et al. (2012) [62]	cross-sectional study three mining sites, Ecuador year not reported gold miners (occupational (direct) Hg exposure): n = 200 (male, mean age 37 years, occupational exposure 0–36 years, mean occupational exposure 9 years, intermittent amalgam burning: 146 burned in last 200 days) gold merchants (occupational (direct) Hg exposure): n = 37 (male, mean age 31 years, occupational exposure 1–14 years, mean occupational exposure 9 years, regular amalgam burning: daily) controls (environmental (non-occupational) Hg exposure): n = 72 (male, mean age 38 years)	laboratory: blood sample: Hg spot urine sample: Hg, creatinine examination: questionnaire: socio-demographics, lifestyle, occupation, Hg exposure neurological examination: postural tremor, hand coordination, reaction time, postural stability	laboratory: significantly elevated levels in miners in comparison to controls: U–Hg + P–Hg (no difference in B–Hg) B–Hg: merchants >>> miners > controls 11% miners + 71.4% merchants exceed BEI limit (>15 µg/L) miners: 0.7–100 µg/L (mean 5.3 µg/L) P–Hg (no detailed numbers reported): merchants >>> miners > controls U–Hg: merchants >>> miners > controls (miners: depends very much on time since last burning amalgam) 5.1% miners + 61.1% merchants exceed BEI limit (>35 µg/L) miners: 0.3–170 µg/gCr (mean 3.3 µg/gCr)	associations: significantly positive: B–Hg: − tremor (centre frequency left) **(r_s_ = 0.1)** − reaction time **(r_s_ = 0.2)** U–Hg: − tremor (centre frequency) **(r_s_ = 0.1, r_s_ = 0.1)** − reaction time **(r_s_ = 0.2)** significantly negative: B–Hg: postural sway (velocity) **(r_s_ = −0.2)** U–Hg: postural sway (velocity) **(r_s_ = −0.2)**
Schutzmeier et al. (2016) [63]	cross-sectional study Ecuador Portovelo Zaruma August 2015 gold miners (occupational (direct) Hg exposure): n = 865 (age 18–65 years) participants for a pharmaceutical study: n = 44 (male, age 19–59 years, mean age 38.6 years, work duration ½-40 years, mean work duration 11.2 years, U–Hg ≥ 15 µg/L)	laboratory: spot urine sample: Hg examination: clinical examination drug screening (alcohol, other drugs) diagnosis of intoxication: examination + laboratory parameters	U–Hg (laboratory): gold miners (n = 865): <0.5–163 µg/L (median 1.8 µg/L) 78.3% < HMB-I (7 µg/L), 15.8% 7–25 µg/L, 5.9% >HBM-II (25 µg/L), 3.4% >BAT (35 µg/L) participants (n = 44): 15–163 µg/L (median 35 µg/L) examination (n = 44): medical score sum (most common symptoms (n = 36)): neuro-psychological symptoms (sleeping problems 91.7%, dysdiadochokinesia 88.9%, ataxia of gait 72.2%, tremor 13.9%) problems in neuro-psychological tests (matchbox test 91.7%, heel-to-shin test 88.9%, pencil tapping test 75%) other symptoms: discoloured gum 33.3%, salivation 33.3%, proteinuria 2.8% most frequent other symptoms (n = 24): physical + mental fatigue 62.5%, social problems (54.2%), irritability 54.2%	correlations: significantly positive (weak/moderate): U–Hg + medical score sum **(r_s_ = 0.4)**

**Table 3 ijerph-19-02081-t003:** Bias assessment. (XS: cross sectional study).

Study	Study Design	Internal Validity—Bias	Internal Validity—Confounder (Selection Bias)	Performance Bias	Detection Bias	Attrition Bias	Reporting Bias	Others
Afrifa et al. (2017) [43]	XS	low risk	low risk	low risk	low risk	low risk	low risk	contradictory data (The text (page 6) reports a different odds ratio for serum creatinine than the corresponding table (page 7).)
Afrifa et al. (2018) [51]	XS	low risk	low risk	low risk	low risk	high risk	low risk	none
Rajaee et al. (2015) [44]	XS	high risk	low risk	low risk	high risk	high risk	low risk	none
Mensah et al. (2016) [52]	XS	high risk	high risk	low risk	low risk	low risk	low risk	none
Bose-O’Reilly et al. (2010a) [45]	XS	low risk	low risk	low risk	low risk	low risk	low risk	none
Harada et al. (1999) [53]	XS	high risk	high risk	low risk	high risk	high risk	low risk	(1) H–Hg as general marker for Hg exposure (2) MeHg values: n = 9
Bose-O’Reilly et al. (2008) [46]	XS	low risk	low risk	low risk	low risk	low risk	low risk	none
Steckling et al. (2014) [54]	XS	low risk	high risk	low risk	high risk	low risk	low risk	only the study in Zimbabwe could be taken into account
Tayrab (2017) [55]	XS	high risk	high risk	low risk	low risk	high risk	low risk	none
Tomicic et al. (2011) [56]	XS	high risk	high risk	low risk	high risk	high risk	low risk	none
Wanyana et al. (2020) [57]	XS	low risk	low risk	low risk	high risk	low risk	low risk	contradictory data (The text says that 75.8% of the miners use PPE (page 5), while the table says that 75.8% of the miners do not use PPE (Table 2).)
Bose-O’Reilly et al. (2010b) [47]	XS	low risk	low risk	low risk	low risk	low risk	low risk	none
Ekawanti and Krisnayanti (2015) [58]	XS	high risk	high risk	low risk	high risk	high risk	low risk	no detailed information about child labour in the control group
Khan et al. (2012) [59]	XS	high risk	high risk	low risk	high risk	high risk	low risk	(1) contradictory data (The text (page 2) gives the concentrations of T–Hg in RBC and plasma for children, who work as miners, in another order than the table (Table 1).) (2) missing data (There is no number of participants reported (n=unknown).)
Riaz et al. (2016) [60]	XS	high risk	high risk	low risk	low risk	high risk	low risk	contradictory data (The text says that the values of T–Hg in hair are higher for female miners, although this is—according to the reported data—not the case.)
Lacerda et al. (2020) [61]	XS	high risk	low risk	low risk	high risk	high risk	low risk	(1) H–Hg as general marker for Hg exposure (2) contradictory data (The text (page 2) reports different ages for the gold miners and the riverines than Table 1.)
Branches et al. (1993) [48]	Case series	high risk	high risk	low risk	high risk	high risk	low risk	contradictory data (The number of urban inhabitants differs in the text (n = 21 on page 4, n = 22 on page 8).)
Harari et al. (2012) [62]	XS	high risk	high risk	low risk	low risk	low risk	low risk	none
Schutzmeier et al. (2016) [63]	XS	low risk	low risk	low risk	high risk	low risk	low risk	none

## Data Availability

Data can be requested by the authors of this review.

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
