# Peer review of "Mercury Exposure and Its Health Effects in Workers in the Artisanal and Small-Scale Gold Mining (ASGM) Sector—A Systematic Review"

_ijerph, 2022, doi:10.3390/ijerph19042081_

Round 1

Reviewer 1 Report

Overall Comments:

The paper is too long and repeats should be reduced. The authors have made an attempt to analyse suitable literature on the ill effects of mercury overdose on employees in the field of ASGM. The discussion needs to be more scholarly and include some clear derivations of the analysis and compare with animal studies or other correlation studies that may be in the literature. This will add credibility to the work and provide better evidence that mercury overdose does cause ill health and a push to improve working conditions for these employees.

Specific Comments

Introduction should include some data on the scale of the mercury poisoning, what is the burden of this to human health and environment health worldwide and regionally

Line 70-73:

This can be better explained: are the authors saying the while many processes cause mercury emissions including natural processes the greatest contributor is ASGM?

Line 78-79: explain this better “in experimental settings, the absorption rate of vapour amounts to 67 – 87% in humans” I think the authors are explaining in that 67-87 % is absorbed via inhalation. If so it has to be better explained

Line 94-95 meaning is not clear, if this is related to which part of the body it can be found then it is not clear

Line 215- 216 this eligibility criteria is not properly explained

Figure 1 inclusion criteria should be in a table or as footnote to figure 1 , the text does not explain it clearly

Line 377-379 what does “this industry” refer to ASGM? Please make clear

The crux of this paper is in the results are explained in table 2, in order to reduce the article length reduce any duplicate explanation in text of what is already mentioned in table 2

Conclusion:

Are there any lab animal studies of mercury poisoning that can be used to compare or substantiate the conclusions. While the limitations are clearly laid out, the significance of the findings and their application for future research and intervention should be better written

Author Response

The paper is too long and repeats should be reduced. The authors have made an attempt to analyse suitable literature on the ill effects of mercury overdose on employees in the field of ASGM. The discussion needs to be more scholarly and include some clear derivations of the analysis and compare with animal studies or other correlation studies that may be in the literature. This will add credibility to the work and provide better evidence that mercury overdose does cause ill health and a push to improve working conditions for these employees.

Answer: We thank the reviewer for this valuable remark. We screened the whole manuscript carefully and deleted sentences and paragraphs in  the introduction as well as in the results and discussion section in order to reduce the length of the manuscript.  In addition, we added a paragraph concerning toxicology studies in animals, their main findings and expected health consequences in humans. We also   highlighted the need of improved working conditions for miners in the “Conclusions”.

Specific Comments

Introduction should include some data on the scale of the mercury poisoning, what is the burden of this to human health and environment health worldwide and regionally

Answer: We thank the reviewer for this suggestion and included another reference in our paper, dealing with the worldwide and regionally burden of disease regarding mercury exposure among workers in the Introduction section.

Line 70-73: This can be better explained: are the authors saying the while many processes cause mercury emissions including natural processes the greatest contributor is ASGM?

Answer: We thank the reviewer for this note. We added a short explanation to clarify the contribution of natural and anthropogenic sources of atmospheric mercury emissions.

Line 78-79: explain this better “in experimental settings, the absorption rate of vapour amounts to 67 – 87% in humans” I think the authors are explaining in that 67-87 % is absorbed via inhalation. If so it has to be better explained

Answer: We thank the reviewer for this note and restructured the sentence to clarify the proportion of absorbed mercury.

Line 94-95 meaning is not clear, if this is related to which part of the body it can be found then it is not clear

Answer: In order to reduce the length of the manuscript and with regard to the scope of the manuscript, we omitted a paragraph regarding HBM values and summarized that the internal burden can be quantified using human biomonitoring, preferably by using blood or urine samples.

Line 215- 216 this eligibility criteria is not properly explained

Figure 1 inclusion criteria should be in a table or as footnote to figure 1 , the text does not explain it clearly

Answer: We thank the reviewer for this note and added a table (Table 1) in the methods section to illustrate our inclusion and exclusion criteria.

Line 377-379 what does “this industry” refer to ASGM? Please make clear

Answer: The production of fluorescent lamps also exposes workers to mercury and may cause mercury-related symptoms. We added an explanation to make this point clear.

The crux of this paper is in the results are explained in table 2, in order to reduce the article length reduce any duplicate explanation in text of what is already mentioned in table 2

Answer: We thank the reviewer for this remark. We critically looked thorough our paper and shortened it wherever possible. We deleted some paragraphs in the introduction and discussion section. in addition, we tried to shortened the results part in the text. In accordance to the other reviewers, we restructured the Table 2 for a better understanding.

Conclusion:

Are there any lab animal studies of mercury poisoning that can be used to compare or substantiate the conclusions. While the limitations are clearly laid out, the significance of the findings and their application for future research and intervention should be better written

Answer: We thank the reviewer for this remark. We added a paragraph to underline the significance of our systematic review – the lack of causal studies of mercury exposure and its health effects in ASG miners.

Reviewer 2 Report

Comments on ijerph-1584475

This study tries to review exposure and health effects of mercury in primitive gold mining. Since adverse effects of mercury in gold mining are serious especially in the developing nations, it is important to summarize current research and global trend. In this regards, I think this manuscript may be worth publishing, but needs to be revised before accept.

Similar problems are also reported in China, Mongolia, South Africa, and other countries, but haven’t any surveys been reported in those countries?

Table 2 is extremely difficult to understand for readers. Parameters should be categorized and make some simple and readable tables. If the contents of data extraction and bias assessment are different, they should be present in separate tables. If most of the citing studies are cross-sectional, you can summarize cross-sectional studies in a table and others can be explained in main text. If data is complicated to be shown in a table, you can explain the trends as figures.

Studies on inorganic mercury and methylmercury should be discussed separately because the source, kinetic, outcome are completely different. I think the problem in gold mining could be attributed to inorganic mercury, but why do you review also organic mercury?

Author Response

This study tries to review exposure and health effects of mercury in primitive gold mining. Since adverse effects of mercury in gold mining are serious especially in the developing nations, it is important to summarize current research and global trend. In this regard, I think this manuscript may be worth publishing, but needs to be revised before accept.

Similar problems are also reported in China, Mongolia, South Africa, and other countries, but haven’t any surveys been reported in those countries?

Answer: We thank the reviewer for the overall positive evaluation and are grateful to discuss the items, the reviewer mentioned. The reviewer is absolutely correct, from the more than 10’000 studies we used for the screening process, also studies from China, Mongolia and other countries were initially available. Unfortunately, many studies did not meet our inclusion criteria. Most studies that were excluded used an ecological study design and detected differences between an exposed area and an unexposed control area. However, the participants from the exposed area were often not only workers but also residents. Since we wanted to focus on direct occupational and not solely environmental exposure, we had to exclude those studies. We added a brief explanation in the results section “Literature and screening process”

Table 2 is extremely difficult to understand for readers. Parameters should be categorized and make some simple and readable tables. If the contents of data extraction and bias assessment are different, they should be present in separate tables. If most of the citing studies are cross-sectional, you can summarize cross-sectional studies in a table and others can be explained in main text. If data is complicated to be shown in a table, you can explain the trends as figures.

Answer: We thank the reviewer for this helpful suggestion. In order to improve the readability, we merged the columns “measurements” and “examination”. In addition, we separated the tables optically to improve the readability. We also excluded the organic mercury values from the table (see below). Unfortunately, our initial original layout was shifted during the submission process. Hopefully, the format will be adopted for the revised version.

Studies on inorganic mercury and methylmercury should be discussed separately because the source, kinetic, outcome are completely different. I think the problem in gold mining could be attributed to inorganic mercury, but why do you review also organic mercury?

Answer: We thank the reviewer for this remark. The reviewer is absolutely correct that mercury exposure among gold miners is attributed to inorganic mercury. We mentioned organic mercury briefly since two included studies measured mercury concentrations in hair. However, hair mercury is mainly a reliable marker for chronic organic mercury exposure. We discussed the impaired quality of mercury measurements in the discussion section. Furthermore, we excluded measurements of organic mercury concentrations from our table to focus more on our topic of inorganic mercury exposure. However, according to another reviewer we included a short paragraph highlighting the different mercury compounds.

Reviewer 3 Report

I have one concern that I would like to see the authors address.  I feel the authors glossed over the profound differences in toxicity of mercury.  Organo mercury is simply way more toxic than elemental mercury.  for example, the authors state: "However, mercury accumulation occurs in organ tissues, as for example kidneys, thyroid or brain, but its exact distribution depends on its chemical form [25,26]."  which is not only true, but also extremely important in terms of how a toxic response manifests.  SO, I would like to see a bit more in the introduction about how wildly different toxic responses can be based upon the state of mercury (organic, inorganic, or elemental).  Furthermore, I think it would be really beneficial to bring up in the discussion why it is important to study the species of mercury (e.g. specific mercury compounds) in these various mining practices.  This might tell us more about where we should focus our intervetions.

In the end, Hg is nasty stuff...really, really bad...and many of these 'artisinal' miners are at great risk for overexposure, regardless of the chemistry, but the toxic dose and organ systems affected are highly dependent on the form of Hg.  That point is made...but is not RESOUNDINGLY clear.  I think it should be as it might explain the variance observed in the studies.

Overall, nice paper.

Author Response

I have one concern that I would like to see the authors address.  I feel the authors glossed over the profound differences in toxicity of mercury.  Organo mercury is simply way more toxic than elemental mercury.  for example, the authors state: "However, mercury accumulation occurs in organ tissues, as for example kidneys, thyroid or brain, but its exact distribution depends on its chemical form [25,26]."  which is not only true, but also extremely important in terms of how a toxic response manifests.  SO, I would like to see a bit more in the introduction about how wildly different toxic responses can be based upon the state of mercury (organic, inorganic, or elemental).

We thank the reviewer for this suggestion. In accordance to another reviewer, who suggested us to focus on inorganic mercury, we reformulated our paragraph in the introduction section concerning the different mercury forms and their effects on human health and highlighted in particular the toxicokinetics of elemental mercury. We added more information about the pathway from inhalation to target organ, in particular the brain with its corresponding neuro-psychological abnormalities.

Furthermore, I think it would be really beneficial to bring up in the discussion why it is important to study the species of mercury (e.g. specific mercury compounds) in these various mining practices.  This might tell us more about where we should focus our intervetions.

We thank the reviewer for this remark. We added a short paragraph concerning organic vs inorganic mercury in the discussion section “Data extraction”.

In the end, Hg is nasty stuff...really, really bad...and many of these 'artisanal' miners are at great risk for overexposure, regardless of the chemistry, but the toxic dose and organ systems affected are highly dependent on the form of Hg.  That point is made...but is not RESOUNDINGLY clear.  I think it should be as it might explain the variance observed in the studies.

We thank the reviewer for this suggestion. We added a sentence concerning the variance and a possible explanation regarding the different mercury compounds in the discussion section.

Overall, nice paper.

Answer: Thank you very much, we really appreciate your valuable input.